# Distributed sensing of earthquakes and ocean-solid Earth interactions on seafloor telecom cables

A. Sladen [1]*, D. Rivet[1], J. P Ampuero[1], L. De Barros [1], Y. Hello [1], G. Calbris [2] & P. Lamare [3]

Two thirds of the surface of our planet are covered by water and are still poorly instrumented, which has prevented the earth science community from addressing numerous key scientific questions. The potential to leverage the existing fiber optic seafloor telecom cables that criss-cross the oceans, by using them as dense arrays of seismo-acoustic sensors, remains to be evaluated. Here, we report Distributed Acoustic Sensing measurements on a 41.5 km-long telecom cable that is deployed offshore Toulon, France. Our observations demonstrate the capability to monitor with unprecedented details the ocean-solid earth interactions from the coast to the abyssal plain, in addition to regional seismicity (e.g., a magnitude 1.9 micro-earthquake located 100 km away) with signal characteristics comparable to those of a coastal seismic station.

[1] Université Côte d'Azur, CNRS, Observatoire de la Côte d'Azur, IRD, Géoazur, 250 rue Albert Einstein, Valbonne 06560, France. [2] Febus-optics, Technopole Helioparc - 2 av, Président Pierre Angot - 64 000, Pau, France. [3] Aix Marseille Université, CNRS/IN2P3, CPPM, 163, avenue de Luminy, Marseille 13288, France. *email: sladen@geoazur.unice.fr

About 70% of the Earth's surface is covered by oceans, thus barely accessible to in situ instrumentation and opaque to remote sensing. Paradoxically, our vision of the geologic and biologic richness of the oceans remains fragmentary. The challenge of instrumenting the oceans is critical as it holds the answers to numerous fundamental scientific questions, such as the dynamics of the oceans, the internal structure of the Earth, and the complex interaction between life, geology, and oceans. This challenge also encompasses the monitoring of various natural resources and natural hazards (earthquakes, tsunamis, submarine landslides), including those in coastal areas that are increasingly vulnerable in a changing climate.

Adapting instrumentation to the extreme conditions of the ocean floor (pressure, biofouling, corrosion) requires expertize and is costly, with the main hurdle being the cost of ship time for deployment and recovery. Drifting sensors, such as the Argo and Mermaid floats[1,2], can rapidly cover large areas, but remain limited by satellite transmission, power supply, and poor control over sensor locations. Permanent seafloor observatories for long-term monitoring[3], comprised of multi-physics platforms connected to land by an electro-optic cable, are very costly to install and maintain[4], which limits their spatial extent, density, and scope. Although these different approaches have enabled significant discoveries, our observations below the oceans' surface and sea-bottom remain limited.

Owing to the rise of the internet, most oceans are crossed by fiber optic cables that present the possibility to leverage this infrastructure for scientific purposes. Recognizing this opportunity, an International Joint Task Force was established in 2012 to design "SMART" (Scientific Monitoring And Reliable Telecommunications) cables with environmental sensors embedded in repeater boxes placed every ~50 km[5]. Recently, the capacity to turn optical fibers (OF) into seismo-acoustic sensors has been developed. One breakthrough was the coincidental discovery that earthquakes can be detected by analyzing the phase stability of state-of-the-art lasers across thousand-kilometer-long seafloor telecommunication cables[6]. Yet, this approach only provides one measurement, which is integrated over the entire length of the cable. An alternative approach, called Distributed Acoustic Sensing (DAS), exploits the phase of light that is backscattered by the inherent inhomogeneities of the silica fiber to provide densely spaced, high-rate measurements of strain. DAS can provide high frequency (1 kHz) acoustic measurements with metric spacing, effectively turning OF cables into dense linear seismic arrays[7].

The technology has been used in the oil and gas industry for many years now, but has only recently revealed its full potential for seismological and environmental applications[8–13]. DAS on seafloor telecom cables is not as straightforward as it might initially seem, because these are sturdy cables with jelly compound layers and metallic armoring. These barriers are designed to provide stability and protection from torsion and external disruptions, most notably by fishing activities in coastal areas[14]. In addition, weak coupling between the cable and the poorly consolidated sediments on the seafloor may prevent accurate recording of seismic waves. Notwithstanding strong expectations[15] and one reported earthquake detection[16], the performance of DAS on submarine telecom cables remains to be evaluated to better define its range of possible applications.

Here, we present results of DAS measurements performed on a telecom-type cable. We show that the measurements are indeed highly sensitive to seismo-acoustic signals, and that such cables can readily provide continuous, dense measurements over large distances to study a range of marine and solid earth phenomena with unprecedented levels of detail.

## Results

**The experimental setup.** Data were acquired from February 19th to February 24th, 2018, on a 41.5 km long electro-optic telecommunication cable from Alcatel, which is deployed offshore Toulon in the south of France. This cable is the backbone of the MEUST-NUMerEnv project[17] (Mediterranean Eurocentre for Underwater Sciences and Technologies—Neutrino Mer Environnement) (Fig. 1). The cable straddles several oceanic domains of the north Mediterranean margin: a shallow continental shelf, a steep continental slope, and a 2500 m-deep oceanic plain. The main purpose of the cable is to collect data from the KM3NeT/ORCA (Oscillation Research with Cosmics in the Abyss) neutrino detector deployed at its seafloor termination point[18]. The MEUST cable was installed in 2014[19] and, as it is often the case with telecommunication cable, was simply laid on the seafloor. The cable is only buried for the first 2 km offshore (Supplementary Note 1).

The DAS interrogator unit is connected to one end of the fiber. Coherent pulses of light are emitted through the OF and phase changes of the backscattered light signal are continuously recorded. Local extension and contraction between two locations of the fiber induced by environmental seismic perturbation cause

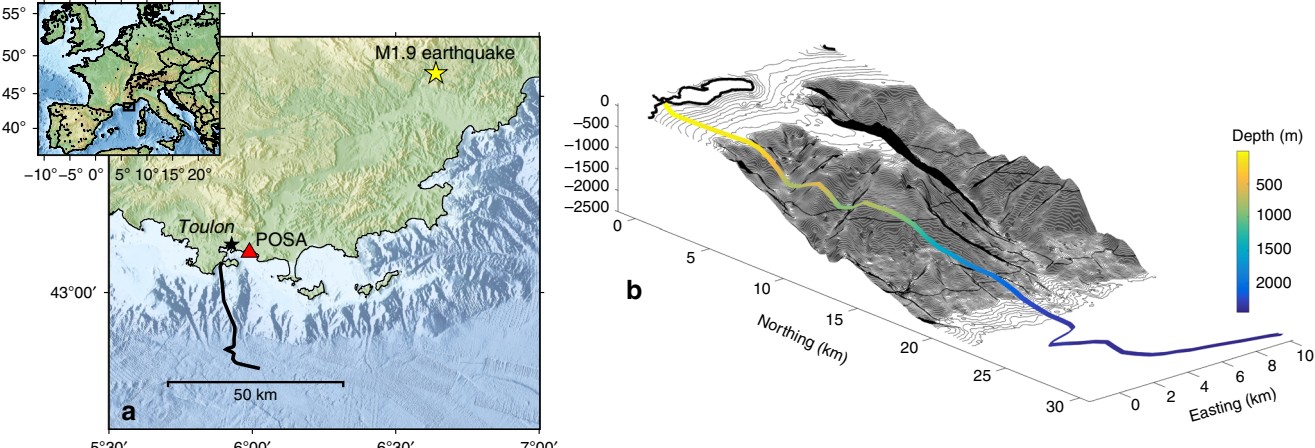

**Fig. 1 Map and perspective view of the seafloor MEUST-NUMerEnv cable.** The optic fiber cable offshore Toulon, France, shown in **a** map view and **b** 3D view. The 41.5 km long optical fiber crosses several oceanic domains: the shallow continental shelf, the steep continental slope and the deep oceanic plain. The yellow star on the map indicates the location of a magnitude 1.9 earthquake. The red triangle is the permanent seismic station POSA.

linear phase changes of the reflected backscattered signals allowing to measure associated strain or strain-rate in the longitudinal direction[20]. The distance between two phase change measurements is called the gauge length and a measurement consists in the integration of the perturbation along the gauge length. In most cases, a better signal to noise ratio is obtained for a longer gauge length but it needs to stay smaller than the shorter wavelengths targeted. In our experiment, the gauge length was fixed to 19.2 m. The data set used in this study consists of strain-rate value sampled every 6.4 m in space and 0.5 ms in time yielding ~ 6500 acoustic sensors with an effective recording frequency of 1 kHz (more details available in the Methods section).

Along its entire length, the OF cable continuously records a background signal made of periodic oscillations. These oscillations have different properties on the continental shelf and on the abyssal plain. We describe them sequentially hereafter, starting with observations near the coast.

**Ocean surface gravity wave detection in the coastal environment.** The signals in the first 8 km of underwater cable are dominated by periodic oscillations with frequencies between 0.1 and 0.25 Hz, which propagate landward with increasing amplitude (Fig. 2b, c). Their amplitude decays rapidly with depth as predicted by the linear theory of gravity waves in a water layer of finite thickness $h$:

$$\frac{P_d(h)}{P_{d0}} = \frac{1}{cosh(k \cdot h)} \qquad (1)$$

with $P_d(h)$ and $P_{d0}$ being the dynamic pressure at the seafloor and at the surface, respectively, and $k$ being the wavenumber. The OF cable senses the dynamic pressure produced by the surface gravity wave down to a depth of 100 m, close to the lower limit of wave action, where amplitudes as low as 1 nstrain/s (nano-strain per-second) are measured. The observed depth of extinction of the shallow perturbations depends on their frequency in a manner that is consistent with the linear gravity wave theory (blue dotted curve in Fig. 3e). A frequency-wavenumber analysis (Fig. 2d) reveals that the signal is mainly composed of two dispersive wave trains, propagating landward and cross-shore with speeds decreasing from ~ 25–10 m/s. The dispersion curve associated with the strongest amplitudes is effectively explained by the dispersion relationship of the linear gravity wave theory[21]:

$$\omega = \sqrt{g \cdot k \cdot tanh(k \cdot h)} \qquad (2)$$

where $g$ is the acceleration of gravity and $\omega$ the frequency. For the comparison in Fig. 2d, the depth $h$ is fixed to 100 m, because this is the average depth for the first 8 km of the cable. The second dispersion curve appears to be similar to the first one but is compressed along the $k$-axis and therefore can be explained as a

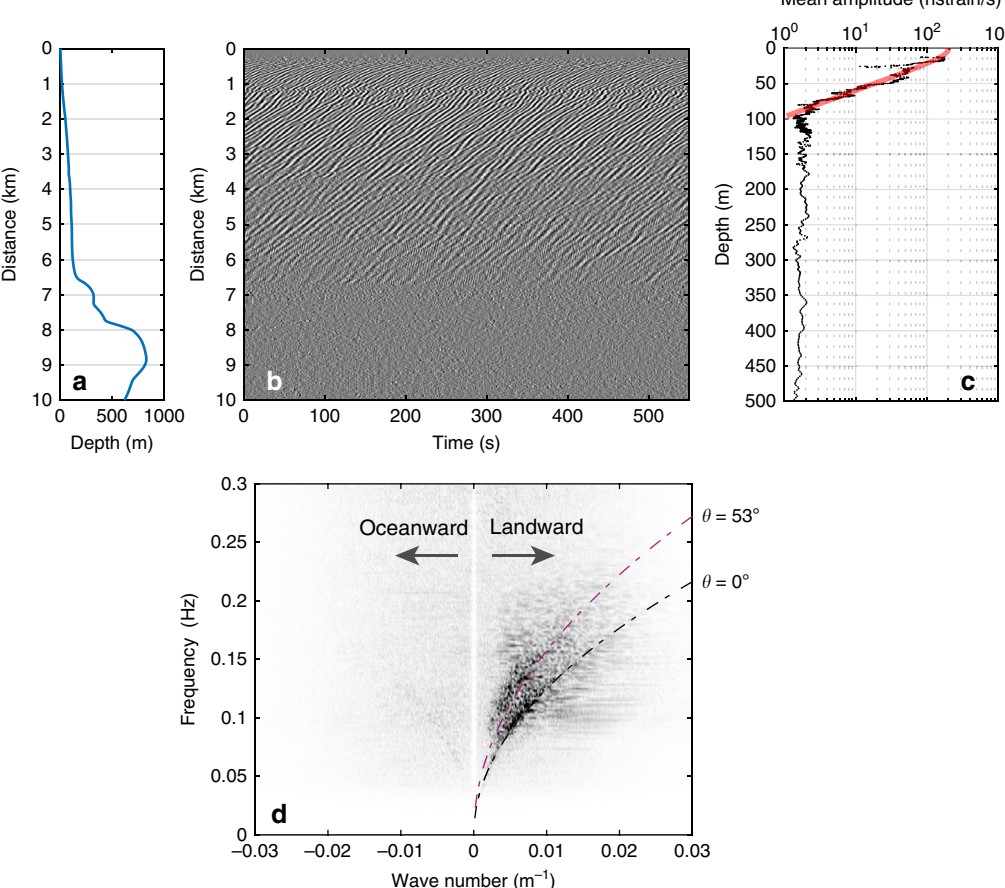

**Fig. 2 Seismic seafloor signal caused by oceanic surface gravity waves. a** Depth profile and **b** 550 s long record of strain-rate along the first 10 km of the cable. Each trace is normalized by its maximum amplitude. The data show periodic oscillations mainly propagating towards the shore. **c** Mean strain-rate over the same distance as a function of depth and theoretical prediction for intermediate depth regime and a wavelength of 100 m. **d** Frequency-wavenumber *f–k* decomposition of the strain-rate signal, for seaward (*k* < 0) and landward (*k* > 0) components, and linear gravity wave dispersion curves for two different incidence angles assuming a water depth of 100 m (dashed curves).

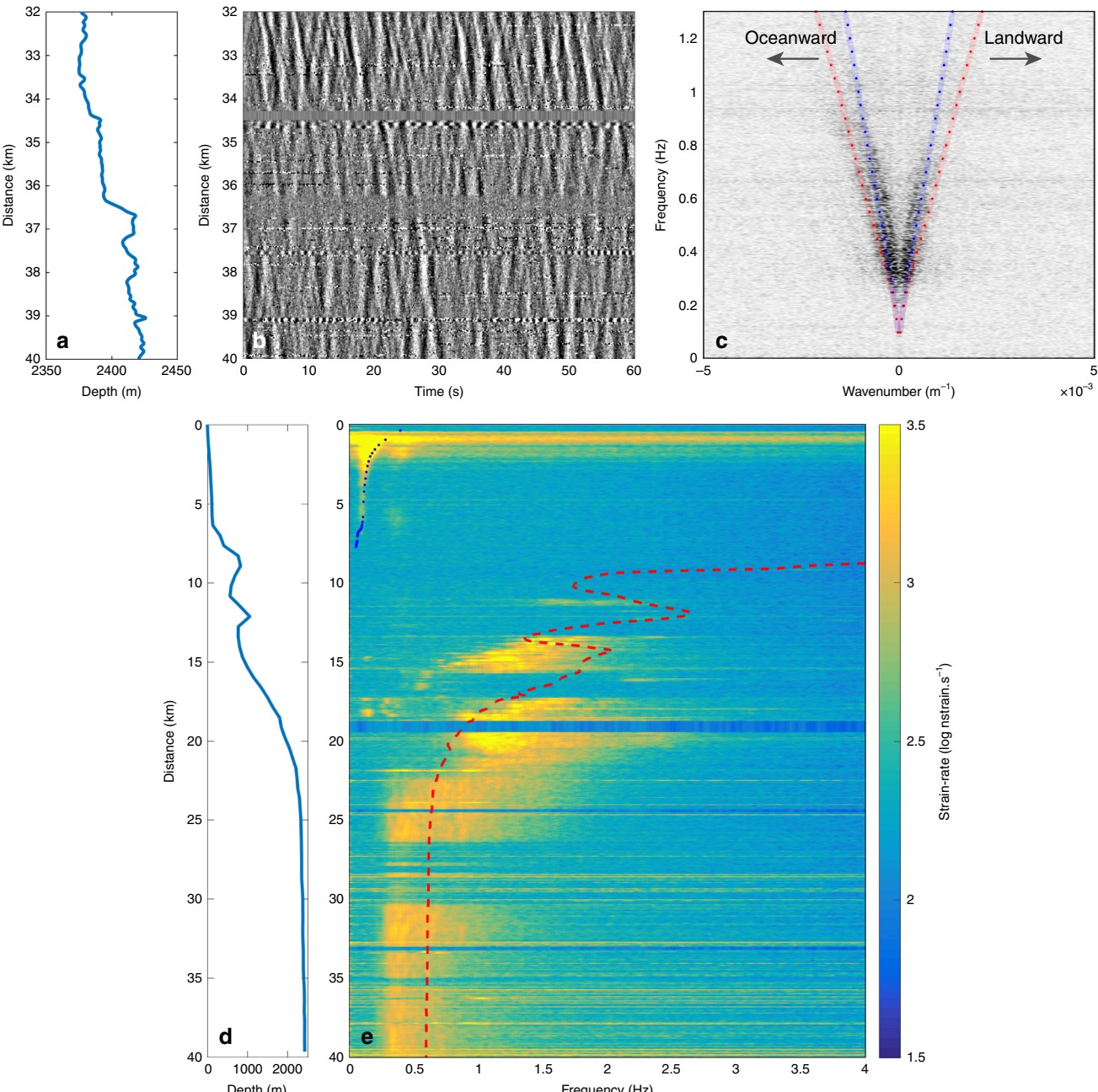

**Fig. 3 Observations of oceanic secondary microseismic noise. a** Depth profile of the OF cable. **b** 60 s record of strain-rate between km 32 and 40. **c** *f–k* decomposition of the signal. The dotted lines are the modeled frequencies of peak amplitude (minimum group velocity) of the fundamental mode of Scholte waves assuming a regional velocity model[26] (Supplementary Fig. 2) and varying the P-wave velocity in the top sedimentary layer (1500 m/s in red and 2000 m/s in blue). **d** Depth profile of the OF cable along its full length. **e** Spectrum of the noise along the cable. The pressure directly produced by the swell vanishes at a depth that depends on frequency as predicted by linear theory (blue dotted curve shows predicted depth of 95% amplitude loss). Beyond 8 km from the coast, at depths larger than 200 m, the OF cable senses the second-order pressure fluctuations caused by the sea surface waves. In deep water, the frequencies of maximum energy are consistent with those predicted from the water column resonance effect that amplifies the Scholte waves (red dashed curve).

wave train impinging on the cable at a different angle, near 53 degrees. Besides the two dominant wave trains, we detect a third signal propagating oceanward with much smaller amplitudes. Because its dispersion curve is symmetric to that of the main landward wavefield, we infer that it corresponds to a reflection off the coast. Its faint amplitudes are coherent with field experiments showing that most (> 90%) of the sea-swell energy is dissipated in the surf zone[22]. At depths > 100 m, the background amplitude becomes independent of depth (Fig. 2c) as predicted by the theory

of second-order pressure fluctuation[23]. Overall, the OF cable provides direct and continuous observations of seafloor perturbations owing to the interaction between the shoaling bathymetry and the ocean waves[24] (Supplementary Movie 1). This interaction is one of the major sources of micro-vibrations traveling through the solid earth, and is known as primary microseismic noise.

**Secondary microseism noise generation in the abyssal plain.** In the abyssal plain, below a depth of 2000 m, the background signal

is dominated by two opposite wave trains propagating with similar dispersion properties (Fig. 3a, b). Compared with the continental shelf records, these higher frequency waves (mainly 0.2–0.8 Hz) propagate at speeds close to that of acoustic waves in water (1500 m/s), suggesting that they are Scholte waves, surface waves propagating along the seafloor interface (Fig. 3b). This interpretation is confirmed by simulations of the fundamental Scholte mode[25] using a regional 1D velocity model[26] (Fig. 3b). The speed of Scholte waves is mainly controlled by the velocity of the top sedimentary layer and changing the compressional value to 1500 or 2000 m/s is sufficient to explain the slight spreading of the dispersion curves (Fig. 3c). These waves are thought to result from the interaction of ocean waves propagating in opposite directions creating second-order pressure fluctuations. These fluctuations oscillate at double the ocean wave frequency and have been shown to couple into seismo-acoustic waves down to the seafloor[23]. Indeed, the dominant frequency of the swell observed on the continental shelf is between 0.1 and 0.25 Hz (Fig. 2d), whereas the dominant frequency observed in the abyssal plain is between 0.2 and 0.5 Hz (Fig. 3e). This opposite wave interaction is the main contributor to the secondary microseismic peak identified in the noise spectrum of worldwide broadband seismic stations[27]. The dominant frequency of the Scholte waves changes with depth as a result of the water column resonance effect that amplifies certain frequencies (Fig. 3d, e)[24].

The direct observation of these two major sources of seismic noise—wave–wave and wave-bathymetry interactions—demonstrates the potential of DAS to study the underlying physical processes and to quantify their spatio-temporal evolution. The dense monitoring of seafloor pressure fluctuations and interface seismic waves could also serve as a proxy to monitor the evolution of oceanic waves[28]. The acoustic excitation of the seafloor caused by surface oceanic waves leads to the primary and secondary microseismic peaks in the noise spectrum. To our knowledge, this is the first time that the evolution of the ocean-solid earth interactions have been tracked continuously from the coast to the deep ocean.

**Detection of a M1.9 regional earthquake.** DAS is now frequently used in the oil industry to monitor seismicity associated to a wellbore with the optical fiber cable in vertical position[29]. One earthquake, magnitude 3.9, was recorded using a seafloor fiber optic cable south of Toyohashi, Japan[16]. The P- and S-waves of the earthquake, located ~100 km away, were identified. The region of our experiment is characterized by more moderate seismic activity. Over the duration of the experiment, the regional seismic network detected several earthquakes with magnitudes between 1 and 2.3 (http://sismoazur.oca.eu/). The POSA broadband seismic station, part of the regional network, is located close to the cable landing site and provided reference measurements (Fig. 1). The largest event clearly recorded by the POSA station over the time period of the study was an earthquake of local magnitude 1.9 that occurred on 20 February 2018 at 05:31:32 UTM, north-east of the cable at a distance of 80 to 100 km (lat: 43.581˚, lon: 6.635˚, Fig. 1). The wavefield recorded by the OF is dominated by S- and surface waves, which is consistent with the POSA record in which the P-waves are barely visible (Fig. 4). Hence, the very small amplitude of the P-wave on the OF is owing to wave attenuation and source radiation pattern, and not to a lack of OF sensitivity. Similar seismic phases were therefore observed with the OF and an on-land broadband station (Supplementary Fig. 1).

On both sensors, the earthquake can be seen in the same frequency band, between 2 and 20 Hz (Supplementary Fig. 1), and its signal is buried in noise at lower and higher frequencies. The noise spectrum also appears similar on both sensors. For such a small earthquake, the peak ground velocity in POSA only reaches 2 µm/s. Under the assumption of an incoming plane wave, strain can be multiplied by the apparent velocity (here 2155 m/s) and compared with the POSA ground velocity signal[8]. This results in a signal twice as large, which is consistent with amplification by the presence of soft sediments on the seafloor (Supplementary Fig. 1). Therefore, we find that the OF cable is able to record very small earthquakes with a sensitivity comparable to that of a broadband seismic land station.

**Variable cable coupling to the seafloor.** There was a significant variability in the amplitude of the recorded strain-rate along the entire cable length with two clear minima ~ 8 and 14 km (Fig. 4). These minima correspond to sections where the cable straddles canyons and probably does not directly lie on the seafloor, thus the coupling is poor. This interpretation would also explain the stronger coupling observed on the edges of the canyons where the cable has to support the extra weight of its neighboring sections. Alternatively, the sections of higher amplitude could correspond to site effects with amplification of the seismic wavefield at the bathymetry highs. Similarly, high amplitude signals from 17–20 km are likely related to freely oscillating portions of the OF in an area of steep and rough bathymetry. It is not clear which hypothesis to favor given the uncertainties on the exact location of the cable and on the morphology and composition of the seafloor. The weak signal observed between 28 and 33 km could be attributed to a local change in the cable orientation due to repair operations in 2017 but the stronger signal over small sections suggests that it is more likely owing to a lower coupling, possibly related to the redeployment of the cable on top of the sediment cover following the repair operations. Beyond these different fluctuations, we notice a slight increase in the mean amplitude response of the cable, by ~15%, beyond 18 km (Supplementary Fig. 3). This point corresponds to the transition from armored to non-armored cable structure. In summary, the signal quality and ground coupling over the 41.5 km of cable seem to be mainly controlled by variations in the bathymetry and sediment cover, rather than changes in the cable structure: the transitions of the cable from double to simple armoring, or to lightweight protection are not associated with drastic changes in the quality of the recordings. These questions related to the coupling will need to be explored further in order to take full advantage of the measurements. Years of lab and field testing in land seismic applications provide a valuable background information to interpret these sensitivity variations[30,31]. Yet, two inherent limitations of seafloor cables are that the cable ships have little control on the way the cable is laid on the seafloor, and the sediment cover is likely to vary spatially and through time.

## Discussion

The results presented in this study demonstrate that DAS is directly applicable to seafloor telecommunication cables. Owing to its dense spatial and temporal sampling of seismo-acoustic signals, in the oceans and along their margins, DAS promises a wide spectrum of scientific and environmental applications. Here, we have presented records related to a small local earthquake, ocean surface gravity wave and microseismic noise, but the approach could be applied to the monitoring of many other sources of acoustic signals, such as those generated by mammals or marine traffic.

DAS is based on reflectometry, meaning that the signal analyzed is inherently weaker and has a shorter range than the direct signal. At present, attenuation limits the range of DAS systems to ~50 km on a standard optical fiber. This range already opens the

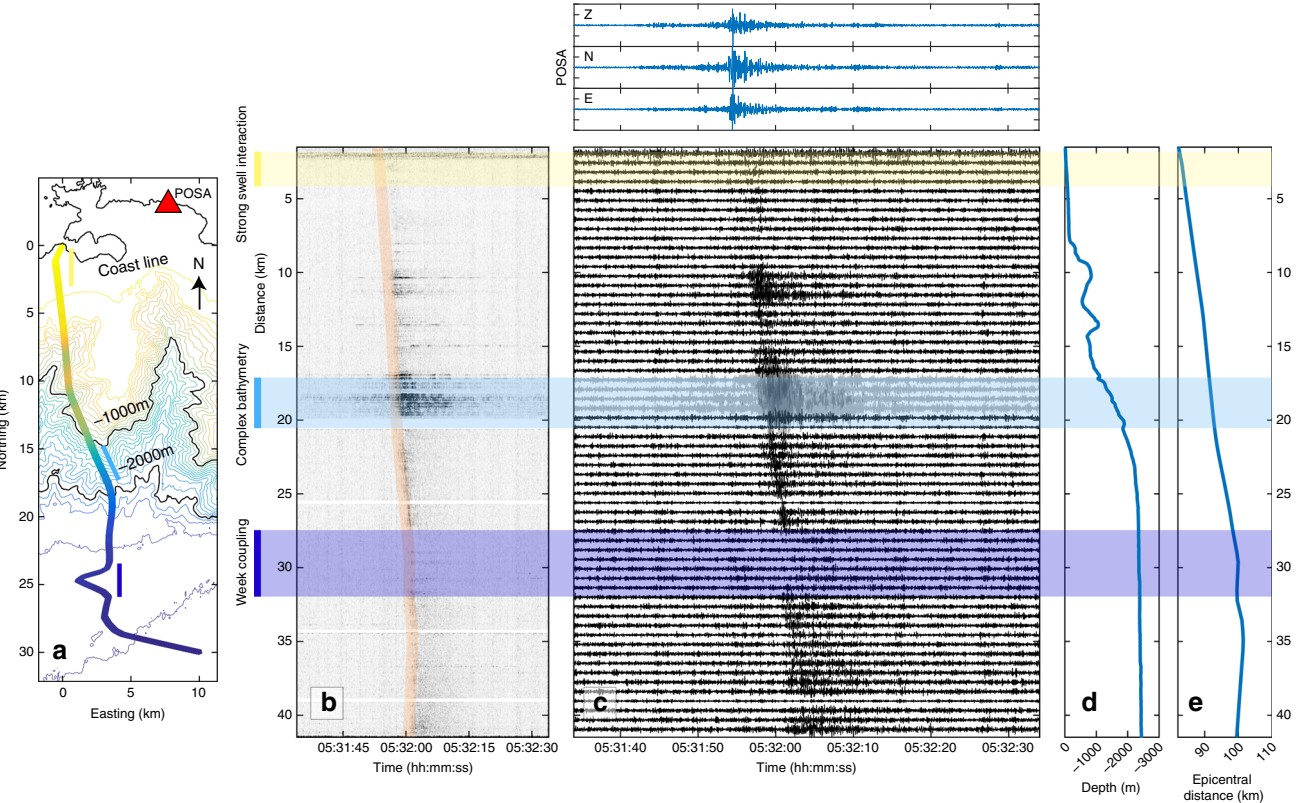

**Fig. 4 Comparison of the M1.9 earthquake recorded at land station POSA and along the fiber optic cable. a** Bathymetry map of the OF cable and location of the broadband land station POSA (red triangle). DAS data are plotted either as strain-rate absolute amplitude filtered between 1 and 15 Hz measured every 10 m **b**, or as strain-rate averaged along 320 m sections of the cable **c**. The shaded portion in **c** indicates the section of cable with high amplifications and where the OF cable crosses a complex bathymetry. On top are plotted for comparison the three-component records of the on-land seismic station POSA. The two plots on the right side show the cable depth and epicentral distance at different distances along the cable.

door to many applications, such as the monitoring of active and passive margins, from the coast to the abyssal plains, thus encompassing most marine and geologic processes (e.g., subduction earthquakes, landslides, coastal erosion processes), including those of greater socio-economic concern. To counteract the effect of attenuation, telecommunication cables are equipped with uni-directional amplifiers every 30–50 km, which would prevent the back-scattering of light and DAS measurements beyond that point. Our laboratory tests indicate that it is possible and easy to extend the range of the measurements beyond 50 km using a bi-directional amplifier instead, and without any significant loss in sensitivity (Supplementary Fig. 4). The range might even be extended beyond 300 km following the same approach[32]. Discussion with cable manufacturers will have to take place to assess if this modification can be implemented on trans-oceanic telecommunication cables and at reasonable cost. But we believe that our results are strong arguments of its scientific and socio-economic interest. This modification would only apply to new cables but rapid global coverage opens an opportunity. Indeed, cable routes are expanding worldwide to address the rapidly growing demand of the internet, and most existing cables were installed in the mid-2000's and will have to be replaced in the next decade.

DAS has several key advantages over transmission analysis: it only requires access to one end of the seafloor cable (transmission analysis would not have been possible on the MEUST-NUMerEnv cable) and it provides dense spatial sampling of the wavefield. This latter advantage enables an additional range of applications, such as array techniques for detecting and tracking the sources of seismo-acoustic signals[33,34], reconstructing the

water wave elevation[35], or passive imaging techniques to infer the internal structure of the ocean and of the Earth[8]. Array techniques using communication cables that are already in place along subduction margins could allow near-field tracking of the seismic rupture and more-effective seismic and tsunami early warning systems[36,37]. With the long lifetime of seafloor communication cables, typically designed to operate 25 years, imagery of the seafloor could be repeated over time to monitor changes in its properties, like those related to coastal erosion, fluid circulation, or extraction. The possibility to monitor the dynamics of the seafloor beyond just a few sampling points now appears to be within reach.

## Methods
**Characterics of the DAS system and processing parameters**. The DAS system used for the experiment is a commercial Febus A1 interrogator developed by Febus Optics company. The system relies on a pending patent single-pulse architecture and an optical heterodyne detection to extract the phase of the light[38].

The most important parameter when performing strain measurement using phase sensitive DAS system is the gauge length and we set it at 20 m. the pulse width only needs to be smaller than half the gauge length[39]. Since this rule is respected, no major effect is observed when varying the pulse width. In our case, to optimize the signal to noise ratio, the pulse width was chosen to be half the gauge length.

Optical power has been adjusted such that the optical amplitude signal is strong enough to correctly extract the phase until 41 Km while preventing any non-linear effects when light propagate along the optical fiber. Non-linear effects would engender distortion on the optical signal and thus on the strain measurement.

## Data availability
The fiber optic DAS recordings of the earthquake and microseism signals are available in the following OSF repository: https://osf.io/x6awb/.

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

## Acknowledgements

We thank the team from the Centre de Physique des Particules de Marseille who facilitated the access to the MEUST infrastructure. The MEUST infrastructure is financed with the support of the CNRS/IN2P3, the Region Sud, France (CPER the State (DRRT), and the Europe (FEDER). This work was supported by the SEAFOOD project, funded in part by grant ANR-17-CE04-0007 of the French Agence Nationale de la Recherche and in part by Université Côte d'Azur IDEX program UCA<sup>JEDI</sup> ANR-15-IDEX-0001.

## Author contributions

A.S. designed the experiment and wrote the manuscript. D.R. performed most of the data processing. A.S, D.R. J-P.-A., L. de B., and G.C. contributed to the signal interpretation and discussion of the results. G.C. performed the bi-directional amplification laboratory test. Y.H., G.C., and P.L. supported the operational and organizational aspects in Toulon.

## Competing interests

G.C. is employed by Febus Optics S.A. the company developing the DAS system used for the acquisition of the data analyzed in this study. There is a pending patent on the DAS system developed by Febus Optics S.A.
