## [Peer Review File · Nature Communications]

Reviewers' comments:

Reviewer #1 (Remarks to the Author):

The authors report on seismo-acoustic measurements over a 41.5 km-long underwater optical fiber cable using distributed acoustic sensing equipment. A small local earthquake and ocean-solid earth interaction were detected with extremely high spatial resolution and detail. Particularly interesting is the measured evolution of the ocean-solid earth interaction from the coast to deep ocean.

The manuscript is well written, present a solid and novel body of work, providing further proof that optical fibers are a powerful emerging tool for Earth Sciences. The data obtained from DAS measurements can be beneficial to a number of science areas, so I think the manuscript has a good multi-disciplinary appeal. For all these reasons I believe it deserves to be published in this journal.

However, as the technique is relatively new and the reader might not be familiar with it, I would strongly recommend that the authors not only show the advantages of the technique but also make its limitations clearer. As seafloor telecommunication cables crossing the oceans are mentioned, it should be clearer from the main text that only the coastal sections of these cables can be used with DAS and not their entire length. The authors imply this in the Supplementary Materials by showing extension of the DAS range using repeaters, but this should be apparent already in the main text.

The authors state that "with small modifications the repeaters installed all along the seafloor telecommunication cables could re-amplify the signal to reach greater distances". In my opinion the authors make a very weak case in this instance. As described by the authors themselves in the manuscript introduction, the prohibitive cost of installing permanent seafloor instrumentation has, to-date, prevented the implementation of a network of ocean bottom sensors (such as ocean bottom seismometers), leaving a large fraction of the Earth's surface uncovered. Modifying existing repeaters, as the author suggest, faces similar challenges, both financial and technical and for these reasons is impractical. Fiber-based techniques are only advantageous when the existing underwater infrastructure can be used without modifications.

The term "repeater" is used improperly here. What the authors refer to are bi-directional optical amplifiers and I would suggest to change the term in the manuscript.

On page 2, I believe the sentence "...allowing to measure associated strain or strain-rate in the longitudinal" has not been completed?

If possible, I would suggest to add colours to Fig 4C as it is quite difficult to read as it stands. Perhaps the length intervals the authors refer to in the text (8 to 14 km, 17 to 20 km, 28 to 33 km, etc..) could be shown in different colours for better clarity.

Reviewer #2 (Remarks to the Author):

The paper's main claims are to use existing subsea infrastructure for monitoring seismic and wave-seabed interaction using distributed acoustic sensing (DAS). This is distinct, as pointed out by the authors, from the earlier work by Marra et al. (2018) that they cite, in that Marra measured the integral strain (as derived from the phase of the light emerging from the fibre) along a telecommunications whereas the present work is distributed (i.e. it resolves the wave). By and large the paper is well written, although there are some Gallicisms that need to be addressed (such as the use of "dilatation" when the authors probably mean "extension"). However, I do not think that the paper meets the threshold for novelty and significance specified in the

review criteria. It seems to me not to meet the criteria for novelty so one then considers the advance over available materials and I feel that this case is also not made in the paper. I also noted that [5] is not about optical cables: it is about taking the opportunity of legacy coaxial cables being replaced by optical cable to re-purpose the old cables for connecting to electrical gauges or perhaps making electro-magnetic measurements using these cables.

The use of existing subsea cables for a number of environmental applications including monitoring of seismicity was discussed and reviewed in some detail in A. H. Hartog, M. Belal, and M. A. Clare, "Advances in Distributed Fiber-Optic Sensing for Monitoring Marine Infrastructure, Measuring the Deep Ocean, and Quantifying the Risks Posed by Seafloor Hazards," *Marine Technology Society Journal*, vol. 52, no. 5, pp. 58-73, 2018. The concept as discussed in the present manuscript was actually demonstrated and actually demonstrated by Kimura, T., Araki, E., & Yokobiki, T. 2018. "Submarine earthquake event recording while wave monitoring using submarine optical fiber cable and DAS Technology". In: Japanese Geophysical Union Meeting 2018. Chiba, Japan. STT50-04. Admittedly, the range achieved in the latter paper was substantially shorter than is presented here, but nonetheless, it does anticipate the work in this manuscript. The Kimura et al. work actually shows clear waves in the time/distance domain.

The strength of the manuscript is the more detailed analysis and interpretation of the data than I have seen elsewhere in particular, the effect of waves (gravity waves) on the results. Comments on the effect of armouring are also of interest. That might be of interest as regards tsunami warnings although that has also been proposed in another paper by Kimura ("Potential for real-time tsunami monitoring using DAS technology". In: JpGU-AGU Joint Meeting 2017. Chiba City, Japan. Tokyo, Japan: JpGU (Japanese Geophysical Union). HDS16-P01.). On the other hand, if one is really interested in the earthquakes themselves, the coupling to the seabed is paramount – the movement of waves above the cable is of secondary importance. So it may be that not all subsea are suitable for this type of work and that only those that are trenched can be used reliably. The process by which the pressure of the gravity waves is transferred to the optical fibres within the cable was not clear to me. In general, such cables are designed to isolate the fibres from strain on the cable but we also know that acoustic vibrations are still transferred to the fibres in some cases. If the fibres are in loose tubes within the, would the frictional forces be sufficient to strain the fibres even at the very low frequencies discussed in the paper? However, if we really are discussing changes in hydrostatic pressure, how is that transferred to the fibres? Generally, submarine optical telecommunication cables protect the fibres in thin steel tubes that provide a barrier to pressure, as well as to water and hydrogen ingress. So how do the authors think that the pressure is transferred? These uncertainties weaken the "strength of evidence" criteria given to reviewers to judge the relevance of the manuscripts.

The paper also shows scant detail as to the optics of the DAS system that is employed. Are they actually using the technique of Posey et al [17] or some other form of DAS? There are many forms of DAS and they each have their strengths and limitations. I am not sure if the authors are from an instrument developer or a team of earth scientists but assuming that they did not just buy a DAS instrument and apply it, there should be more information as to the techniques used to extract the phase, the optical power, the pulse duration and all the aspects that allow the sensitivity of the system to be judged. This makes it difficult to reproduce the work for another researcher in the same field.

In summary, I think that the work is certainly of interest to the scientific community, but that *Nature Communications* is not a suitable journal: there is insufficient novelty and the detail that would bring out the value of the paper is not here. So my recommendation would be to move the paper to a full-length journal where the novelty is less of an issue and then to strengthen the paper providing a full description of the optics (unless it has been published elsewhere) and to provide a more detailed analysis of the signals and in particular of the cable-seabed coupling

Reviewer #3 (Remarks to the Author):

Dear Authors,

I fully enjoyed reviewing your manuscript. It present interesting and new observations and is clearly written. I am including an annotated PDF file with a few suggestions. Most of them are minor and including them in the final version of the manuscript would be quick. My only technical question is: why did you choose to compare the DAS data with the E component of the POSA seismometer? Given the configuration of the experiment, and the DAS directivity, it would have made more sense to compare the DAS data with the POSA N component. In addition, it seems that the N component has recorded a stronger P wave arrival than the E component. In the annotated PDF I also suggested to include a couple of earlier references that present earthquakes and surface waves recorded by using DAS systems interrogating an onshore "dark fiber" cable.

Best regards,

Biondo Biondi

Comments from authors are highlighted in blue below and in the updated version of the manuscript.

Reviewer #1 (Remarks to the Author):

The authors report on seismo-acoustic measurements over a 41.5 km-long underwater optical fiber cable using distributed acoustic sensing equipment. A small local earthquake and ocean-solid earth interaction were detected with extremely high spatial resolution and detail. Particularly interesting is the measured evolution of the ocean-solid earth interaction from the coast to deep ocean.

The manuscript is well written, present a solid and novel body of work, providing further proof that optical fibers are a powerful emerging tool for Earth Sciences. The data obtained from DAS measurements can be beneficial to a number of science areas, so I think the manuscript has a good multi-disciplinary appeal. For all these reasons I believe it deserves to be published in this journal.

However, as the technique is relatively new and the reader might not be familiar with it, I would strongly recommend that the authors not only show the advantages of the technique but also make its limitations clearer. As seafloor telecommunication cables crossing the oceans are mentioned, it should be clearer from the main text that only the coastal sections of these cables can be used with DAS and not their entire length. The authors imply this in the Supplementary Materials by showing extension of the DAS range using repeaters, but this should be apparent already in the main text.

The authors state that "with small modifications the repeaters installed all along the seafloor telecommunication cables could re-amplify the signal to reach greater distances". In my opinion the authors make a very weak case in this instance. As described by the authors themselves in the manuscript introduction, the prohibitive cost of installing permanent seafloor instrumentation has, to-date, prevented the implementation of a network of ocean bottom sensors (such as ocean bottom seismometers), leaving a large fraction of the Earth's surface uncovered. Modifying existing repeaters, as the author suggest, faces similar challenges, both financial and technical and for these reasons is impractical. Fiber-based techniques are only advantageous when the existing underwater infrastructure can be used without modifications.

The penultimate paragraph has been modified to clarify that existing telecommunication cables are currently limited to the section before the first amplifier, so roughly 30 to 50 km from the coast. We now also stress that our results could motivate the cable manufacturers to review the design of these amplifiers, and that these modifications would only apply to future cables. Yet, because of the current worldwide multiplication of cable deployments, and the fact that most cables in place will have to be replaced in the next decade (they were installed in the mid-2000's and their lifetime is about 25 years), the global coverage of most oceans in the near future remains a valid perspective.

The term "repeater" is used improperly here. What the authors refer to are bi-directional optical amplifiers and I would suggest to change the term in the manuscript.

Done. We kept the term "repeater" though in the case of the SMART cable project mentioned in the introduction as it is then referring to the casing and not just the internal optical component.

On page 2, I believe the sentence "...allowing to measure associated strain or strain-rate in the longitudinal" has not been completed?

Done. The word "direction" was missing

If possible, I would suggest to add colours to Fig 4C as it is quite difficult to read as it stands. Perhaps the length intervals the authors refer to in the text (8 to 14 km, 17 to 20 km, 28 to 33 km, etc..) could be shown in different colours for better clarity.

The size of the labels have been increased and semi-transparent coloured strips were added on top of figure 4B to 4D.

Thank you for your time to review our work and your comments.

Reviewer #2 (Remarks to the Author):

The paper's main claims are to use existing subsea infrastructure for monitoring seismic and wave-seabed interaction using distributed acoustic sensing (DAS). This is distinct, as pointed out by the authors, from the earlier work by Marra et al. (2018) that they cite, in that Marra measured the integral strain (as derived from the phase of the light emerging from the fibre) along a telecommunications whereas the present work is distributed (i.e. it resolves the wave).

By and large the paper is well written, although there are some Gallicisms that need to be addressed (such as the use of "dilatation" when the authors probably mean "extension").

Done.

However, I do not think that the paper meets the threshold for novelty and significance specified in the review criteria. It seems to me not to meet the criteria for novelty so one then considers the advance over available materials and I feel that this case is also not made in the paper.

The three references you cite below stress the great potential of applying DAS to underwater cables. Hartog et al. (2018) and Kimura (2018) do not present or analyze DAS signals; only the abstract of Kimura et al. (2018) shows the detection of an earthquake but without any detailed analysis of the signal and they only show a screenshot of the record. To our

knowledge, our work is the first study presenting records of ocean-solid Earth interaction using DAS. Being able to track continuously, densely and over all depth ranges this interaction is well beyond what can be achieved with existing instrumentation. Hence, the approach should motivate a number of future studies in physical oceanography and seismology (origin and characterization of microseisms is fundamental to ambient noise tomography studies). New types of observations usually drive scientific discoveries.

I also noted that [5] is not about optical cables: it is about taking the opportunity of legacy coaxial cables being replaced by optical cable to re-purpose the old cables for connecting to electrical gauges or perhaps making electro-magnetic measurements using these cables. Reference removed and Hartog et al. (2018) introduced a bit further in the text where appropriate.

The use of existing subsea cables for a number of environmental applications including monitoring of seismicity was discussed and reviewed in some detail in A. H. Hartog, M. Belal, and M. A. Clare, "Advances in Distributed Fiber-Optic Sensing for Monitoring Marine Infrastructure, Measuring the Deep Ocean, and Quantifying the Risks Posed by Seafloor Hazards," Marine Technology Society Journal, vol. 52, no. 5, pp. 58-73, 2018. The concept as discussed in the present manuscript was actually demonstrated in And actually demonstrated by Kimura, T., Araki, E., & Yokobiki, T. 2018. "Submarine earthquake event recording while wave monitoring using submarine optical fiber cable and DAS Technology". In: Japanese Geophysical Union Meeting 2018. Chiba, Japan. STT50-04. Admittedly, the range achieved in the latter paper was substantially shorter than is presented here, but nonetheless, it does anticipate the work in this manuscript. The Kimura et al. work actually shows clear waves in the time/distance domain.

The paper of Hartog et al (2018) discusses the potential of the approach but does not show or analyze actual data. As stressed in our manuscript and by your comments on the coupling (see below), it was not clear at all that meaningful signals could be recorded given the specific structure of underwater cables designed to isolate the fibers from external forces.

We were not aware of the abstract of Kimura et al. (2018) who is referring to the detection of an earthquake. It is a one page abstract with limited analysis and only one figure, a screenshot of the earthquake traces. The abstract is now cited.

The strength of the manuscript is the more detailed analysis and interpretation of the data than I have seen elsewhere in particular, the effect of waves (gravity waves) on the results. Comments on the effect of armouring are also of interest. That might be of interest as regards tsunami warnings although that has also been proposed in another paper by Kimura ("Potential for real-time tsunami monitoring using DAS technology". In: JpGU-AGU Joint Meeting 2017. Chiba City, Japan. Tokyo, Japan: JpGU (Japanese Geophysical Union). HDS16-P01.).

This abstract of Kimura (2017) only suggests the idea that DAS could be used on seafloor fiber optic cable but does not show any results. The abstract is now cited in the final paragraph on the perspectives of our work.

On the other hand, if one is really interested in the earthquakes themselves, the coupling to the seabed is paramount – the movement of waves above the cable is of secondary importance. So it may be that not all subsea are suitable for this type of work and that only those that are trenched can be used reliably.

That was our initial worry as the MEUST cable is not trenched. Yet, indeed, we are able to detect various signals, including earthquake signals, with great details. This achievement is one of the main contributions of our work.

The process by which the pressure of the gravity waves is transferred to the optical fibres within the cable was not clear to me. In general, such cables are designed to isolate the fibres from strain on the cable but we also know that acoustic vibrations are still transferred to the fibres in some cases. If the fibres are in loose tubes within the, would the frictional forces be sufficient to strain the fibres even at the very low frequencies discussed in the paper?

It has been demonstrated on land experiments (e.g Lindsey et al., 2017; Jousset et al. 2018) that the fiber is sensing the acoustic wavefield even if in a loose tube. And unless the fiber is surrounded by vacuum, acoustic waves in certain frequency ranges will always reach its core. Here we demonstrate that DAS is sensitive enough to monitor these perturbations.

Lindsey, N.J., Martin, E.R., Dreger, D.S., Freifeld, B., Cole, S., James, S.R., Biondi, B.L., Ajo-Franklin, J.B., 2017. Fiber-Optic Network Observations of Earthquake Wavefields. *Geophysical Research Letters* 44, 11,792-11,799.
<https://doi.org/10.1002/2017GL075722>

Jousset, P., Reinsch, T., Ryberg, T., Blanck, H., Clarke, A., Aghayev, R., Hersir, G.P., Hennings, J., Weber, M., Krawczyk, C.M., 2018. Dynamic strain determination using fibre-optic cables allows imaging of seismological and structural features. *Nature Communications* 9. <https://doi.org/10.1038/s41467-018-04860-y>

However, if we really are discussing changes in hydrostatic pressure, how is that transferred to the fibres? Generally, submarine optical telecommunication cables protect the fibres in thin steel tubes that provide a barrier to pressure, as well as to water and hydrogen ingress. So how do the authors think that the pressure is transferred? These uncertainties weaken the “strength of evidence” criteria given to reviewers to judge the relevance of the manuscripts.

The facts that the signal dispersion curves we obtained are coherent with the theory of ocean surface waves, and that DAS records during an earthquake match nearby seismic land records, leave minimal doubt that the seafloor fiber optic cable is sensitive to these processes.

Our analysis is, for the most part, independent of the quality of the coupling (arrival time of ballistic waves, frequency-wavenumber analysis).

Yet, it is true that the exact process by which the signal is transferred to the fiber is still unclear and will require attention in future studies. Studying how the acoustic signal is filtered by the specific cable structure is beyond the scope of this study.

The paper also shows scant detail as to the optics of the DAS system that is employed. Are they actually using the technique of Posey et al [17] or some other form of DAS? There are many forms of DAS and they each have their strengths and limitations. I am not sure if the authors are from an instrument developer or a team of earth scientists but assuming that they did not just buy a DAS instrument and apply it, there should be more information as to the techniques used to extract the phase, the optical power, the pulse duration and all the aspects that allow the sensitivity of the system to be judged. This makes it difficult to reproduce the work for another researcher in the same field.

The following text has been added to the method section:

Characterics of the DAS system and processing parameters

The DAS system used for the experiment is a commercial Febus A1 interrogator developed by the Febus Optics company. The system relies on a patented single-pulse architecture and an optical heterodyne detection to extract the phase of the light.

The most important parameter when performing strain measurement using phase sensitive DAS systems is the gauge length. We set it at 20 m.

In a previous study, Dean et al. (2016) showed that the pulse width only needs to be smaller than half the gauge length. Since this rule is respected, no major effect is observed when varying the pulse width. In our case, to optimize the signal to noise ratio, the pulse width was chosen to be half the gauge length.

Optical power has been adjusted such that the optical amplitude signal is strong enough to correctly extract the phase until 41 km while preventing any non-linear effects when light propagates along the optical fiber. Non-linear effects would cause distortion on the optical signal and thus on the strain measurement.

Dean, T., Hartog, A., Cuny, T., & Englich, F. (2016). The effects of pulse width on fibre-optic distributed vibration sensing data. In 78th EAGE Conference and Exhibition 2016.

In summary, I think that the work is certainly of interest to the scientific community, but that Nature Communications is not a suitable journal: there is insufficient novelty and the detail that would bring out the value of the paper is not here. So my recommendation would be to move the paper to a full-length journal where the novelty is less of an issue and then to strengthen the paper providing a full description of the optics (unless it has been published elsewhere) and to provide a more detailed analysis of the signals and in particular of the cable-seabed coupling

Thank you for you time to review our work and your comments.

Reviewer #3 (Remarks to the Author):

Dear Authors,

I fully enjoyed reviewing your manuscript. It present interesting and new observations and is clearly written. I am including an annotated PDF file with a few suggestions. Most of them are minor and including them in the final version of the manuscript would be quick. My only technical question is: why did you choose to compare the DAS data with the E component of the POSA seismometer? Given the configuration of the experiment, and the DAS directivity, it would have made more sense to compare the DAS data with the POSA N component. In addition, it seems that the N component has recorded a stronger P wave arrival than the E component.

We corrected the caption as the figure was actually the North component and not East. It absolutely makes more sense as you pointed out.

We looked back at the data, and we confirm that P-waves are of very small amplitudes on all components, and that the first P waves cannot be seen on both horizontal components. We therefore reformulated the text to say that “P waves can be barely seen on all components”.

In the annotated PDF I also suggested to include a couple of earlier references that present earthquakes and surface waves recorded by using DAS systems interrogating an onshore “dark fiber” cable.

The suggested modifications have all been taken into account, and the references added.

Thank you for you time to review our work and your comments.

Best regards,

Biondo Biondi

Reviewers' comments:

Reviewer #1 (Remarks to the Author):

I believe the authors have addressed most of the points raised by the reviewers and have improved the manuscript.

However, the authors describe the replacement of uni-directional amplifier with bi-directional amplifier as "technically simple modification", as we we learn from the telecommunications community, but it is well known that this is far from being a simple task.

Whilst narrow-bandwidth bidirectional amplification is feasible over long fiber links employing several amplifiers, this is very hard to achieve over the whole telecommunication bandwidth as the amplified Rayleigh back-scatter makes the transmission unstable. I would suggest to remove the "technically simple" term and I would also separate the technical challenge from cost considerations.

"Discussion with cable manufacturer will have to take place to see if this modification can be implemented on trans-oceanic telecommunication cables AND at reasonable cost."

Reviewer #2 (Remarks to the Author):

Comments on the response to the reviewers

The authors have addressed some of the raised by me and the other reviewers.

On the subject of prior work, there is now some reference to the earlier work of others, but the statement "the performance of DAS on submarine telecom cables therefore remains to be demonstrated." (on page 2) is contradicted later on by the authors inclusion of a reference to the work of Kimura that clearly does demonstrate detection of earthquakes on submarine communications (in both the Kimura work and the present manuscript, these are dedicated cables linking remote subsea observation equipment back to the mainland). I believe that the present manuscript stands on its own without the need to minimise the prior contributions of others to the same field.

The question of the acquisition system is addressed in part in the Methods section (from that point of view, the comment on page 3 needs to be amended to read "See Methods and Supplementary material for more details..."). I was unable to find the patents (which I assume were filed by Febus Optics) referred to by the authors – a search of the relevant databases shows several Febus patents but none on DAS, so the authors should provide the appropriate reference either to a patent or to an article in the literature, whether by the equipment suppliers or some other source. Similar techniques (heterodyne detection) have been described in the literature, but it is not clear to the reader what distinguishes the equipment used in this experiment from that which has already been described. The reference to Dean et al. (2016) has not been included in the bibliography, as far as I can see.

The new reference to the oil industry (page 5) is incomplete in that borehole seismic measurements are conducted using a variety of configurations, not all cemented (e.g. attached to the outside production tubing or on cables deployed temporarily within the wellbore). This is described in at least one of the references cited already.

Regarding the comments on optical amplification, I fully support the first reviewer: the concept of an optical repeater is completely different from that of a bi-directional optical amplifier. The issues raised have been largely addressed in the new text (page 6). However, I am very sceptical as to the likelihood of optical subsea transmission being adapted to allow for seismic experiments: this is an issue for the cable operators rather than the cable manufacturers. It relates to economics and system reliability and in my view for additional scientific capability to be added to a submarine cable, it will need to be financed by research funding and most likely provided on dedicated fibres. I think that there is a better likelihood of adapting existing research communications infrastructure because at least it is all within the research community and it is not carrying fee-paying traffic.

It should be pointed out that the concept of remote amplification of DAS signals is not particularly new (see for example T. Parker, S. Shatalin, and M. Farhadiroushan, "Distributed Acoustic Sensing – a new tool for seismic applications," *First Break*, vol. 32, no. February 2014, pp. 61-9, 2014, which shows similar data on similar distances) so I am not sure what Fig. S4 adds to this paper. I apologise to the authors and the Editor for not raising this on my first pass through the paper. Regarding the coupling of acoustic waves to the fibre, the authors are of course right – unless they are suspended in vacuum, there will always be some coupling to the fibres. However, the sensitivity could vary by quite a few (3-4) orders of magnitude depending on the details of the coupling. Moreover, the nature of the modulation transferred to the optical backscatter could vary significantly between say a pure pressure wave conveyed by a fluid to the bare fibre and the longitudinal strain applied via the cable structure if coupled to the ground (or seabed) and if the fibre is strain-coupled to the cable. It has taken many years to understand this in the context of land seismic, e.g.

P. Lumens, A. Franzen, K. Hornman, S. G. Karama, G. Hemink, B. Kuvshinov, J. La Follett, B. Wyker, and P. Zwartjes, "Cable development for Distributed Geophysical Sensing, with a field trial in surface seismic," *Fifth European Workshop on Optical Fibre Sensors*, 2013, pp. 879435-879435-5.,

J. C. Hornman, "Field trial of seismic recording using distributed acoustic sensing with broadside sensitive fibre-optic cables," *Geophys. Prosp.*, vol. 65, no. 1, pp. 35-46, 2016.,

B. N. Kuvshinov, "Interaction of helically wound fibre-optic cables with plane seismic waves," *Geophys. Prosp.*, vol. 64, no. 3, pp. 671-688, 2016.

B. Papp, D. Donno, J. E. Martin, and A. H. Hartog, "A study of the geophysical response of distributed fibre optic acoustic sensors through laboratory-scale experiments," *Geophys. Prosp.*, vol. 65, no. 5, pp. 1186-1204, 2016.

Some of these lessons from the literature will be applicable in the present context, but I suspect that much more needs to be learned for interactions at the seabed.

Overall, I think that the strength of the paper is in the oceanography and the interpretation of the results, as mentioned in my initial review and on that basis, the paper should be published. Nonetheless, I feel that the aspects surrounding that central part, such as the optics and acquisition need to be accurate in their statements (e.g. as to novelty) and adequate in their referencing and description.

Reviewer #3 (Remarks to the Author):

Thanks for following my suggestions.

Congratulations for the nice research results and well-written paper.

Comments from authors are highlighted in blue below. In the updated version of the manuscript, the new modifications are in **bold blue**.

Reviewer #1 (Remarks to the Author):

I believe the authors have addressed most of the points raised by the reviewers and have improved the manuscript.

However, the authors describe the replacement of uni-directional amplifier with bi-directional amplifier as "technically simple modification", as we learn from the telecommunications community, but it is well known that this is far from being a simple task.

Whilst narrow-bandwidth bidirectional amplification is feasible over long fiber links employing several amplifiers, this is very hard to achieve over the whole telecommunication bandwidth as the amplified Rayleigh back-scatter makes the transmission unstable. I would suggest to remove the "technically simple" term and I would also separate the technical challenge from cost considerations.

"Discussion with cable manufacturer will have to take place to see if this modification can be implemented on trans-oceanic telecommunication cables AND at reasonable cost."

The sentence has been modified.

Reviewer #2 (Remarks to the Author):

Comments on the response to the reviewers

The authors have addressed some of the raised by me and the other reviewers.

On the subject of prior work, there is now some reference to the earlier work of others, but the statement "the performance of DAS on submarine telecom cables therefore remains to be demonstrated." (on page 2) is contradicted later on by the authors inclusion of a reference to the work of Kimura that clearly does demonstrate detection of earthquakes on submarine communications (in both the Kimura work and the present manuscript, these are dedicated cables linking remote subsea observation equipment back to the mainland). I believe that the present manuscript stands on its own without the need to minimise the prior contributions of others to the same field.

This sentence of the introduction was modified to also acknowledge the abstract of Kimura et al.:

Notwithstanding strong expectations¹⁵ and one reported earthquake detection¹⁶(Kimura et al) the performance of DAS on submarine telecom cables remains to be evaluated to better define its range of possible applications. Here...

The question of the acquisition system is addressed in part in the Methods section (from that point of view, the comment on page 3 needs to be amended to read “See Methods and Supplementary material for more details...”).

Done

I was unable to find the patents (which I assume were filed by Febus Optics) referred to by the authors – a search of the relevant databases shows several Febus patents but none on DAS, so the authors should provide the appropriate reference either to a patent or to an article in the literature, whether by the equipment suppliers or some other source. Similar techniques (heterodyne detection) have been described in the literature, but it is not clear to the reader what distinguishes the equipment used in this experiment from that which has already been described.

Febus has submitted several patents on Distributed optical fiber sensing, more specifically on Brillouin sensing, in which the architecture of the DAS has already been introduced. The patent specific to the DAS has not been published yet. We added the following reference to a publication related to our DAS:

T. Dean, K. Tertyshnikov, R. Pevzner, G. Calbris, C. Jestin and V. Lanticq, "Experimental Measurement of the Effects of Acquisition Parameters on DAS Data Quality", 81st EAGE Conference and Exhibition, 2019.

The reference to Dean et al. (2016) has not been included in the bibliography, as far as I can see.

The reference to Dean et al. (2016) was moved from the Method section to the main list of references.

The new reference to the oil industry (page 5) is incomplete in that borehole seismic measurements are conducted using a variety of configurations, not all cemented (e.g. attached to the outside production tubing or on cables deployed temporarily within the wellbore). This is described in at least one of the references cited already.

We removed the reference to the cemented configuration. Indeed, the cable is often clamped on the tubing.

Regarding the comments on optical amplification, I fully support the first reviewer: the concept of an optical repeater is completely different from that of a bi-directional optical amplifier. The issues raised have been largely addressed in the new text (page 6). However, I am very sceptical as to the likelihood of optical subsea transmission being adapted to allow for seismic experiments: this is an issue for the cable operators rather than the cable manufacturers. It relates to economics and system reliability and in my view for additional

scientific capability to be added to a submarine cable, it will need to be financed by research funding and most likely provided on dedicated fibres. I think that there is a better likelihood of adapting existing research communications infrastructure because at least it is all within the research community and it is not carrying fee-paying traffic.

The sentence was modified following the suggestion of reviewer #1.

We believe that the high number of potential applications of seafloor DAS, matching high economical and societal stakes (earthquake and tsunami early-warning, meteorology, acoustic pollution monitoring ...), could support such developments.

It should be pointed out that the concept of remote amplification of DAS signals is not particularly new (see for example T. Parker, S. Shatalin, and M. Farhadiroushan, "Distributed Acoustic Sensing – a new tool for seismic applications," *First Break*, vol. 32, no. February 2014, pp. 61-9, 2014, which shows similar data on similar distances) so I am not sure what Fig. S4 adds to this paper. I apologise to the authors and the Editor for not raising this on my first pass through the paper.

We agree the concept is not new, but it has not been presented as new in our paper. The paper mentioned by the reviewer is now cited to reinforce the technical feasibility of the concept. As written in that paper, " ... we see no reason why the EDFA chain cannot be increased to extend the range up to the 320 km ...". In our lab tests, we reach 100 km by adding one bidirectional optical amplifier, whereas 3 amplifiers were used in the reference cited to reach 82 km.

Regarding the coupling of acoustic waves to the fibre, the authors are of course right – unless they are suspended in vacuum, there will always be some coupling to the fibres. However, the sensitivity could vary by quite a few (3-4) orders of magnitude depending on the details of the coupling. Moreover, the nature of the modulation transferred to the optical backscatter could vary significantly between say a pure pressure wave conveyed by a fluid to the bare fibre and the longitudinal strain applied via the cable structure if coupled to the ground (or seabed) and if the fibre is strain-coupled to the cable. It has taken many years to understand this in the context of land seismic, e.g.

P. Lumens, A. Franzen, K. Hornman, S. G. Karama, G. Hemink, B. Kuvshinov, J. La Follett, B. Wyker, and P. Zwartjes, "Cable development for Distributed Geophysical Sensing, with a field trial in surface seismic," *Fifth European Workshop on Optical Fibre Sensors*, 2013, pp. 879435-879435-5.,

J. C. Hornman, "Field trial of seismic recording using distributed acoustic sensing with broadside sensitive fibre-optic cables," *Geophys. Prosp.*, vol. 65, no. 1, pp. 35-46, 2016.,

B. N. Kuvshinov, "Interaction of helically wound fibre-optic cables with plane seismic waves," *Geophys. Prosp.*, vol. 64, no. 3, pp. 671-688, 2016.

B. Papp, D. Donno, J. E. Martin, and A. H. Hartog, "A study of the geophysical response of distributed fibre optic acoustic sensors through laboratory-scale experiments," *Geophys. Prosp.*, vol. 65, no. 5, pp. 1186-1204, 2016.

Some of these lessons from the literature will be applicable in the present context, but I suspect that much more needs to be learned for interactions at the seabed.

We've added the following sentences with reference to the papers of Lumens et al. (2013) and Papp et al. (2016):

These questions related to the coupling will need to be explored further in order to take full advantage of the measurements. Years of lab and field testing in land seismic applications provide a valuable background information to interpret these sensitivity variations^{30,31}. Yet, two inherent limitations of seafloor cables are that the cable ships have little control on the way the cable is laid on the seafloor, and the sediment cover is likely to vary spatially and through time.

Overall, I think that the strength of the paper is in the oceanography and the interpretation of the results, as mentioned in my initial review and on that basis, the paper should be published. Nonetheless, I feel that the aspects surrounding that central part, such as the optics and acquisition need to be accurate in their statements (e.g. as to novelty) and adequate in their referencing and description.

Reviewer #3 (Remarks to the Author):

Thanks for following my suggestions.

Congratulations fro the nice research results and well-written paper.

REVIEWERS' COMMENTS:

Reviewer #2 (Remarks to the Author):

The authors have addressed the questions that I raised in the previous iterations and I believe that the manuscript is now in a form suitable for publication.